# Isolation and characterization of the phytopathogenic fungus *Ilyonectria liriodendri* from persimmon as a new susceptible host

Nikolett Molnár[1], Dóra Szabó[1], Adrienn Geiger[1,2], József Geml[1,2], Kálmán Zoltán Váczy[1], Zoltán Karácsony[1]*

1 Food and Wine Research Institute, Eszterházy Károly Catholic University, Eger, Hungary, 2 ELKH-EKKE Lendület Environmental Microbiome Research Group, Eszterházy Károly Catholic University, Eger, Hungary

* karacsony.zoltan@uni-eszterhazy.hu

## Abstract

Several members of the fungal genus *Ilyonectria* primarily infect plants through the roots and basal stem, causing 'black foot' diseases, predominantly in woody plants such as grapevine (*Vitis* spp.) and walnut (*Juglans regia*). In 2021, four *Ilyonectria liriodendri* isolates were cultured from the necrotized roots of *Diospyros virginiana* plants in Eger, Hungary. The isolates were identified by sequencing the ITS, β-tubulin, and partial histone H3 genes. The obtained sequences were used for phylogenetic analysis through multiple sequence alignment and the construction of a Maximum Likelihood tree, which revealed that all four isolates belonged to the species *Ilyonectria liriodendri*. The macro- and micromorphological variations, as well as the differences in exoenzyme production of the isolates suggested that they represent a somewhat diverse set of the same taxon. To prove their association with the symptoms observed in the host plants, the roots of one-year-old *D. virginiana* plants were artificially infected with conidial suspensions of the isolates according to Koch's postulates. After 90 days of incubation in a greenhouse, 16 out of 20 inoculated plants showed necrosis in the taproots, while mock-inoculated plants remained symptomless. Necroses developed in the roots of the infected plants, and the inoculated fungi were reisolated, reinforcing their pathogenicity against *D. virginiana*. To the best of our knowledge, this is the first report of *I. liriodendri* causing disease in persimmon.

## Introduction

The genus *Ilyonectria* belongs to the ascomycetous family Nectriaceae and was described in 2014 [1], with 16 known species to this date (https://www.mycobank.org/). While *Ilyonectria* species are mostly known as phytopathogens, they can grow as saprobes as well as endophytes [2,3]. Most of these species are considered opportunistic soil-borne pathogens of various plants, infecting the roots and the base of the hosts.

**Data availability statement:** All the relevant data are presented in the paper or in the supplementary files.

**Funding:** This work was funded by the National Research, Development and Innovation Office (https://nkfih.gov.hu/english-nkfih) under the project OTKA K-143453. ZK was supported by the János Bolyai Research Scholarship of the Hungarian Academy of Sciences (https://mta.hu/english).

**Competing interests:** The authors declare that they have no known competing financial interests or personal relationships that could have appeared to influence the work reported in this paper.

These pathogens damage the xylem, resulting in "black root rot" and "black foot disease" syndromes [4]. They cause diseases mostly on woody plants like grapevine [5], apple [6], plum, tulip tree [7], loquat [8], olive trees [9], and blackberry [10]. The most widely studied syndrome caused by *Ilyonectria* spp. is the "black foot disease" of grapevine [7,11,12], a member of the so-called grapevine trunk diseases group [13]. Their common symptom is the necrosis of the infected xylem tissues, appearing as black discolorations. The damage to the vascular tissues, probably with the additional effects of fungal phytotoxins [14], results in the decline and the eventual death of the host. While there is no estimation on the economic loss caused by grapevine black foot diseases, both the syndrome and pathogens show a high incidence worldwide [11,15–17].

Persimmon species (*Diospyros* spp.) are also susceptible to vascular infections, causing host decline and significant yield losses [18]. The only bacterial pathogen causing persimmon dieback is *Pseudomonas syringae* [19]. Most of the causal agents are fungal species which also associated with different grapevine trunk disease syndromes. These fungi include *Neofusicoccum* and *Diplodia* species (botryosphaeria dieback), *Eutypa lata* (eutypiosis), *Phaeoacremonium* species (esca disease), *Diaporthe* species (phomopsis disease), as reported by Moyo et al. [20]. The only exception is the ascomycetous *Colletotrichum horii* [21] with no known connection with grapevine trunk diseases. In this context, the lack of fungal species associated with grapevine black foot disease in persimmon is conspicuous.

In 2021, declining young *Diospyros virginiana* plants were investigated in Hungary, showing necrotic spots in their taproots. The fungal species *Ilyonectria liriodendri* was isolated from most of the root samples according to the morphological characteristics and the analysis of ITS, β-tubulin, and partial histone H3 gene sequences of the isolates. To the best of our knowledge, this is the first report of *I. liriodendri* pathogen causing infection in persimmon.

## Materials and methods

### Isolation of fungal strains

Xylem samples were collected from the taproots of 15 declining *D. virginiana* plants in Eger city (Northeastern Hungary, Fig 1), in 2021 from a hobby grower. Five thin discs were cut from each root. The isolation of fungi was carried out as described previously [22]. After the bark tissues were removed, the discs were surface-sterilized in 1% chloramine B solution for 5 min. The sterilized tissues were rinsed in sterile distilled water and dried. Then each sterilized discs were cut into five pieces and placed on potato dextrose agar plates (PDA, Merck KGaA, Darmstadt, Germany). The plates were incubated at room temperature ($21 \pm 2°C$) in the dark for two weeks. Cultures were checked daily, and emerging mycelia were subcultured to new PDA plates to acquire pure cultures for further morphological and molecular identification. A total of 173 fungi were isolated, including 11 *Ilyonectria*-like isolates (from 11 different plants). The *Ilyonectria*-like isolates could be divided into four morphological groups. One representative isolate of each morphological group was subjected to detailed examination.

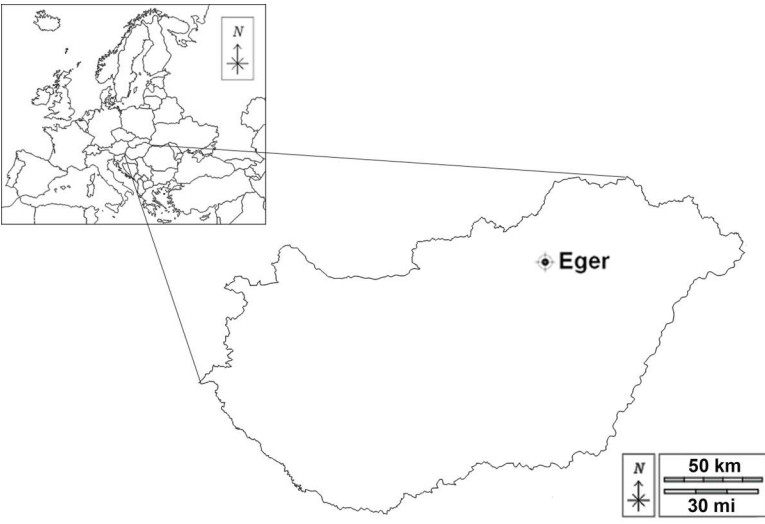

**Fig 1. The origin of fungal strains.** Map showing the location of the fungal isolates obtained in Hungary. The figure was created using online resources (https://d-maps.com/m/europa/hungary/hongrie/hongrie06.gif; https://d-maps.com/m/europa/europemax/europemax10.gif).

## Molecular identification and phylogenetic analyses of fungal isolates

Mycelia from one-week-old PDA cultures were scraped and placed in a microcentrifuge tube. Fungal biomass was lyophilized and ground by the use of sterilized stainless steel bead and Tissuelyzer LT device (Qiagen, Hilden, Germany) at 50 Hz, for 5 min. DNA was extracted by DNeasy Plant Mini Kit (Qiagen, Hilden, Germany) according to the manufacturer's instructions. Polymerase chain reactions (PCRs) were performed to amplify the partial H3 histone gene using CYLH3F (5'-AGGTCCACTGGTGGCAAG-3') and CYLH3R (5'-AGCTGGATG TCCTTGGACTG-3') primers; the ITS region using ITS4 (5'-TCCTCCGCTTATTGATATGC-3') and ITS1F (5'-CTTCGTCATTTAGAGGAAGTAA-3') primers; and the β-tubulin gene using BT2a (5'-GGTAACCAAATCGGTGCTGCTTTC-3') and Bt2b (5'-ACCCTCAGTGTAGTGACCCTTCGC-3') primers as described previously [23]. PCR conditions were 94 °C for 5 min, followed by 35 cycles at 94 °C for 30 s, 52 °C for 30 s and 72 °C for 80 s, and a final elongation at 72 °C for 10 min [24]. The PCR products were visualised on 1% agarose gel. Amplicons were sequenced (BaseClear B.V. -Netherlands) and compared to GenBank using BLAST (www.ncbi.nlm.nih.gov) [25].

For the accurate identification of the examined isolates, the partial H3 histone, the ITS (internal transcribed spacer), and the β-tubulin genes were sequenced. The three previously mentioned loci of 61 *Ilyonectria*, *Dactylonectria* and *Neonectria* strains were collected from the NCBI nucleotide database (https://www.ncbi.nlm.nih.gov/nucleotide/) and used as a reference for the phylogenetic tree construction. Sequences were analyzed on the Phylogeny.fr platform [26] and subsequently aligned with MUSCLE (v3.8.31) [27]. The removal of ambiguous regions was done by Gblocks (v0.91b) using default parameters [28]. A phylogenetic tree was reconstructed according to the maximum likelihood method using the PhyML program (v3.1/3.0 aLRT) [29,30]. Tree visualization was done by TreeDyn (v198.3) [31]. All sequences used in the phylogenetic analysis are listed in Table 1. Identification of the fungal isolates was further verified by species-specific PCR reactions designed to detect *I. liriodendri*, using the primers Cyli F1 (5′-CTC CTC TTC AAC GAT CCG ACG TGC C-3′) and Cyli R1 (5′-GGG GCA GAG CAG ATT TCG-3′), producing a ~200 bp amplicon [6].

## Morphological observations

One cm-wide mycelial disks were cut from the edge of one-week-old colonies growing on PDA and inoculated in the center of 9 cm-wide PDA, malt extract agar (MEA, Merck KGaA, Darmstadt, Germany), oatmeal agar (OA, Merck

Table 1. Sequences used in the phylogenetic analysis. List of fungal strains and GenBank (https://www.ncbi.nlm.nih.gov/genbank/) sequence accession numbers. Strains described in this study are underlined.

| | Strain ID | ITS sequence accession | H3 sequence accession | BTUBULIN sequence accession |
|---|---|---|---|---|
| Ily1 | Ily1 | PV931980 | PP680573.1 | PV943738 |
| Ily3 | Ily3 | PV931981 | PP680574.1 | PV943739 |
| Ily4 | Ily4 | PV931982 | PP680575.1 | PV943740 |
| Ily5 | Ily5 | PV931983 | PP680576.1 | PV943741 |
| Dactylonectria_alcacerensis_1 | BV-1240 | MK602786 | MK579235 | MK602801 |
| Dactylonectria_alcacerensis_2 | Cy133 | JF735331 | JF735628 | JF735459 |
| Dactylonectria_alcacerensis_3 | 129087 | JF735333 | JF735630 | AM419111 |
| Dactylonectria_estremocensis | CBS 112613 | JF735320 | JF735617 | JF735448 |
| Dactylonectria_macrodidyma_1 | EFA446 | MF440371 | MF471474 | MF797794 |
| Dactylonectria_macrodidyma_2 | CBS 112601 | AY677284 | JF735644 | AY677229 |
| Dactylonectria_macrodidyma_3 | CBS 112615 | AY677290 | JF735647 | AY677233 |
| Dactylonectria_macrodidyma_4 | CBS 112594 | AY677282 | JF735643 | AY677231 |
| Dactylonectria_macrodidyma_5 | CBS 112603 | AY677285 | JF735645 | JF735469 |
| Dactylonectria_macrodidyma_6 | CBS 112605 | AY677287 | JF735646 | AY677230 |
| Dactylonectria_macrodidyma_7 | CBS 20709 | JX231163 | JX231147 | JX231115 |
| Dactylonectria_novozelandica_1 | KARE2037 | MK400310 | MK409912 | MK409877 |
| Dactylonectria_peuciseptata_1 | CBS 100819 | EF607090 | JF735582 | EF607067 |
| Dactylonectria_peuciseptata_2 | CBS 120171 | EF607089 | JF735587 | EF607066 |
| Dactylonectria_torresensis_1 | CBS_129086 | NR121500 | JF735681 | JF735492 |
| Dactylonectria_torresensis_2 | CBS 119.41 | JF735349 | JF735657 | JF735478 |
| Dactylonectria_torresensis_3 | CBS 112598 | JF735351 | JF735662 | JF735479 |
| Dactylonectria_vitis | CBS 129082 | JF735303 | JF735580 | JF735431 |
| Ilyonectria_capensis_1 | CBS_132815 | NR152887 | JX231135 | JX231103 |
| Ilyonectria_changbaiensis_4404 | txid2590834 | MF350464 | MF350437 | MF350410 |
| Ilyonectria_communis_1512 | txid2590833 | MF350456 | MF350429 | MF350402 |
| Ilyonectria_coprosmae | CBS 119606 | JF735260 | JF735505 | JF735373 |
| Ilyonectria_crassa_1 | CBS 139.30 | JF735275 | JF735534 | JF735393 |
| Ilyonectria_crassa_2 | CBS 158.31 | JF735276 | JF735535 | JF735394 |
| Ilyonectria_cyclaminicola | CBS 302.93 | JF735304 | JF735581 | JF735432 |
| Ilyonectria_cyclaminicola_1 | CBS_302_93 | NR121495 | JF735581 | JF735432 |
| Ilyonectria_europaea_1 | CBS 129078 | JF735294 | JF735567 | JF735421 |
| Ilyonectria_europaea_2 | CBS 537.92 | EF607079 | JF735568 | EF607064 |
| Ilyonectria_ilicicola_1 | Cy-FO-225 | KY676884 | KY676866 | KY676878 |
| Ilyonectria_leucospermi_1 | CBS_132809 | NR152889 | JX231145 | JX231113 |
| Ilyonectria_liriodendri_1 | CBS_110_81 | NR119565 | JF735507 | DQ178170 |
| Ilyonectria_liriodendri_2 | CBS 117527 | DQ178165 | JF735509 | DQ178172 |
| Ilyonectria_liriodendri_3 | CBS 117526 | DQ178164 | JF735508 | DQ178171 |
| Ilyonectria_liriodendri_4 | MBAE5MY | MT711168 | MT708551 | MT748009 |
| Ilyonectria_liriodendri_5 | MBAE7MY | MT711169 | MT732971 | MT748010 |
| Ilyonectria_liriodendri_6 | MBAE10MY | MT711170 | MT732972 | MT748011 |
| Ilyonectria_liriodendri_7 | CBS 117640 | DQ178166 | JF735510 | DQ178173 |
| Ilyonectria_liriodendri_8 | CBS 112596 | AY677264 | JF735511 | AY677239 |
| Ilyonectria_liriodendri_9 | CBS 112607 | AY677269 | JF735512 | AY677241 |
| Ilyonectria_mors-panacis_1 | CBS 306.35 | JF735288 | JF735557 | JF735414 |
| Ilyonectria_mors-panacis_2 | CBS 124662 | JF735290 | JF735559 | JF735416 |

*(Continued)*

**Table 1.** (Continued)

| | Strain ID | ITS sequence accession | H3 sequence accession | BTUBULIN sequence accession |
|---|---|---|---|---|
| Ilyonectria_protearum_1 | CBS_132812 | NR152890 | JX231149 | JX231117 |
| Ilyonectria_palmarum | DiGeSA-HF7 | HF937432 | HF922621 | HF922609 |
| Ilyonectria_pseudodestructans_2 | CBS 117824 | JF735292 | JF735419 | JF735562 |
| Ilyonectria_pseudodestructans_3 | CBS 129081 | AJ875330 | JF735563 | AM419091 |
| Ilyonectria_qitaiheensis_H309 | txid2590835 | MF350472 | MF350445 | MF350418 |
| Ilyonectria_robusta_1 | CBS_308_35 | NR157427 | JF735518 | JF735377 |
| Ilyonectria_robusta_2 | CBS 129084 | JF735273 | JF735532 | JF735391 |
| Ilyonectria_rufa_1 | CBS 153.37 | AY677271 | JF735540 | AY677251 |
| Ilyonectria_rufa_2 | CBS 640.77 | JF735277 | JF735542 | JF735399 |
| Ilyonectria_vredehoekensis_1 | CBS_132807 | NR152888 | JX231139 | JX231107 |
| Ilyonectria_venezuelensis | CBS 102032 | AM419059 | JF735571 | AY677255 |
| Ilyonectria_zarorii_1 | CPC_37835 | MW114893 | MW119259 | MW119263 |
| Neonectria_ditissima_1 | CBS 226.31 | JF735309 | JF735594 | DQ789869 |
| Neonectria_ditissima_2 | CBS 835.97 | JF735310 | JF735595 | DQ789880 |
| Neonectria_major | CBS 240.29 | JF735308 | JF735593 | DQ789872 |
| Neonectria_neomacrospora | CBS 118984 | JF735311 | JF735598 | DQ789882 |
| Neonectria_ramulariae_1 | CBS 151.29 | JF735313 | JF735602 | JF735438 |
| Neonectria_ramulariae_2 | CBS 182.36 | JF735314 | JF735603 | JF735439 |

CBS, Westerdijk Fungal Biodiversity Institute, Ultrecht, The Netherlands (https://wi.knaw.nl/).

KGaA, Darmstadt, Germany), or 2%m/v water agar (WA) plates. Cultures were incubated at 25°C in the dark, for 14 days. For microscopic observations, fungal biomass was scraped from colonies growing on PDA, mounted in a drop of distilled water, and covered with a coverslip. Microscopic examinations were done by Olympus BX53F2 (Olympus Corporation, Tokyo, Japan) microscope equipped with differential interference contrast (DIC) optics. Photographs were taken by DP47 camera controlled by CellSens Entry software (Olympus Corporation, Tokyo, Japan). Conidial dimensions were determined by image analysis, using a Burker chamber as a reference, by measuring 25 microconidia and 25 macroconidia for each examined strain.

## Examination of exoenzyme production

The activity of digestive enzymes of Ily1, Ily3, Ily4 and Ily5 strains was compared. Czapek dox media (2% w/v sucrose, 0.2% w/v $NaNO_3$, 0.1% w/v $K_2HPO_4$, 0.05% w/v $MgSO_4$, 0.05% w/v KCl, 0.001% w/v $FeSO_4$, 2% w/v agar) supplemented with various substrates of the different enzymes were prepared. Carboxymethylcellulose (1% w/v) was used for cellulase, water-soluble starch (1% w/v) for amylase and ABTS (2mM) for laccase activity detection.

The *I. liriodendri* strains were inoculated onto the above media, as mycelial plugs of 3 mm diameter growing on PDA medium. The effects of the observed digestive enzymes were detected after 7 days of incubation at 25°C. Lugol's staining was carried out for amylase and cellulase detection, followed by the measurement of the radius of halos developed around the colonies. The visualization of laccase activity was observed through the fungal colonies green-blue colorization. Laccase activity was quantified by calculating the decrease in mean pixel intensities of coloured colonies relative to uninoculated media. All the inoculations and measurements were done in triplicate for each strain.

## Softwares

Photographs and microscopic images were processed by Adobe Photoshop CS6 demo version and image analyses were done by Fiji [32]. Statistical comparisons were done by GraphPad Prism 5 software (GraphPad Software, San Diego California USA, www.graphpad.com) using One-way ANOVA with Tukey's post-hoc test.

## Phytopathogenicity tests

One-year-old seed-grown plants of *D. virginiana* were used to evaluate the virulence of fungal isolates. The taproots were cut back to ~5 cm length, and surface sterilized by immersing in 70% v/v ethanol for 5 min. Injured plants were inoculated by incubating their bases for 30 min in conidial suspensions ($10^6$ conidia/mL in distilled water). Mock inoculations were done by placing plants in distilled water. Inoculated plants were potted in a 1:1 mixture of perlite and commercial soil and were grown in a greenhouse for 90 days. Discs were cut 5–10 mm above the inoculation points, photographed and cut into five pieces. These pieces were surface-sterilized in sodium hypochlorite (4%m/v chlorine) for 2 min, then incubated in 70%v/v ethanol for 2 min, dried, and placed on PDA medium. Cultures were incubated at 25°C for a week, then the emerging mycelia were transferred to new PDA plates. Pure cultures were used for DNA extraction and PCRs. To verify the identity of the re-isolated fungi, the ITS, and partial H3 histone gene were amplified and sequenced. All the infections were carried out on five plants for each fungal isolate and mock inoculation.

## Results and discussion

### Isolation and identification of *Ilyonectria liriodendri* from symptomatic persimmon

The four examined declining *D. virginiana* plants showed black spots in the cross sections of the taproots (Fig 2). These spots are located in the tracheae resulting from tissue necrosis and/or vascular occlusion. The distribution of spots was even (Fig 2A), or circular (Fig 2B).

Four fungal strains (IDs: Ily1, Ily3, Ily4 and Ily5) isolated from symptomatic plants showed similar colony morphology as described for *I. liriodendri* [33], although slight differences were observed among the isolates. The abundant presence of 0–1 septate microconidia (average length between 6.08 and 9.17 µm depending on the isolate, S1 Table), and 1–3 septate macroconidia (average length between 11.86 and 35.23 µm depending on the isolate, S1 Table), as well as the lack of chlamydospores was observed in all examined strains. The above traits are also typical of *Ilyonectria* species [1]. Molecular identification of the isolates was done by multi-locus phylogenetic analysis using partial H3 histone, ITS, and β-tubulin gene sequences.

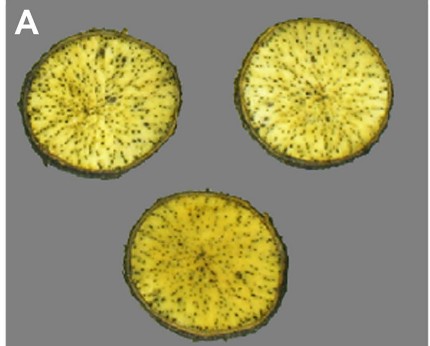 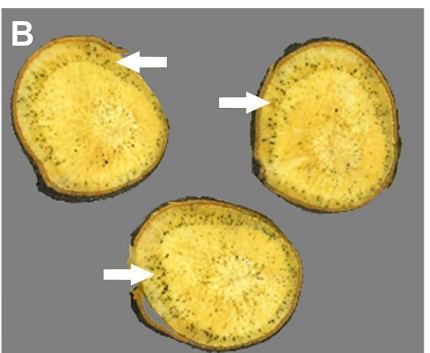 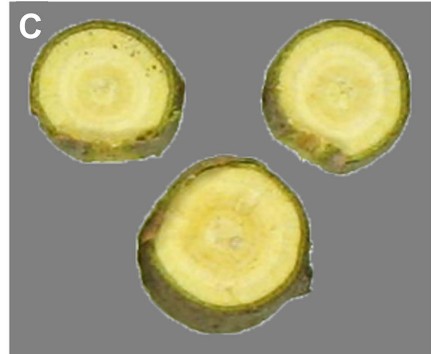

**Fig 2. Symptoms on diseased persimmon plants.** Representative photographs of taproot cross sections of *D. virginiana* plants with even **(A)** or circular **(B)** distribution of necroses, or with asymptomatic roots **(C)**.

## Phylogenetic analyses

The sequences of all four isolates showed the highest similarity to *I. liriodendri* strains based on the three previously mentioned loci and the constructed phylogenetic tree (Fig 3). Despite this similarity, the examined strains formed a separate sister clade with *I. liriodendri* reference strains. To verify that the examined isolates are *I.liriodendri*, species-specific PCR reactions were carried out. The reactions resulted in the expected ~200 bp length amplicon in case of all tested isolates, reinforcing that they belong to *I. liriodendri* (S1 and S2 Figs).

## Morphological characterization of *Ilyonectria liriodendri* isolates

The morphological differences among the four examined isolates suggest that each of them represents a different strain. All isolates formed colonies with even margins on all tested media (Fig 4), but their coloration and relative growth differed. Strain Ily1 formed lemon-yellow colonies on PDA (Fig 4A), Ily4 was pale brown (Fig 4C), while Ily3 (Fig 4B) and Ily5 (Fig 4D) were dark brown on this medium. On MEA all strains showed a lighter coloration (Fig 4E–4H). Smooth, white colony surfaces were observed in the case of Ily1 and Ily4 (Fig 4E and 4G), while Ily3 and Ily5 colonies were both pale yellow and more fluffy (Fig 4F and 4H). Contrary to their coloration, the growth rates of Ily3 and Ily5 greatly differed on MEA medium (1.56 mm/day and 2.86 mm/day respectively, Fig 4F and 4H). Colonies grown on OA showed the similarity of Ily1 with Ily4 and Ily3 with Ily5 (Fig 4I–4L). Colonies of Ily1 and Ily4 were pale brown (Fig 4I and 4K), while Ily3 and Ily5 were dark brown (Fig 4J and 4L). On WA, all isolates formed thin mycelia without significant coloration (Fig 4M–4P), but the lower growth rate of Ily3 (1.24 mm/day) compared to the other strains (1.5–2.1 mm/day) was expressed again, as observed previously on MEA medium.

Microscopic examination of the asexual spores of the *I. liriodendri* isolates (Fig 5) also revealed differences. Micro- and macroconidia were both observed in all strains.

The characterization of conidial morphology (Fig 6 and S1 Table) further reinforced that the four *I. liriodendri* isolates are not clones, but represent a somewhat diverse set of strains. While Ily1 and Ily4 can be barely distinguished according to colony morphology (Fig 4), Ily1 produced significantly shorter microconidia (average length: 6.08 µm, Fig 6A) and macroconidia (average length: 11.86 µm, Fig 6B) compared to any other examined isolate (microconidia average length: 8.17–9.17 µm, macroconidia average length: 29.46–35.23 µm).

## Exoenzyme production of *I. liriodendri* isolates

Secreted degradative enzymes are main pathogenicity factors of phytopathogenic fungi, and their activity usually correlates with virulence. Despite their crucial role in pathogenesis, they show high variance within species, making them suitable for strain characterization. Examination of the digestive exoenzymes revealed further differences among the four *I. liriodendri* strains (Figs 7 and 8 and S2 Table). In the case of laccase activity, the colonies of all strains showed different colorization strength, but the strongest colorization was observable at the Ily5 isolate, which was a significant difference according to the statistical analysis (Figs 7A–7D and 8A). All of the strains produced cellulases according to Lugol's staining of carboxymethylcellulose-containing media, indicated by a clear zone around the colonies (Fig 7E–7H). The Ily 4 and Ily 5 strains, but especially the latter, showed significantly lower cellulase activity than the other strains (Fig 8B). On the starch-containing media, all four strains showed amylase activity (Fig 7I–7L), but in the case of Ily4 strain, this activity was significantly lower (Fig 8C).

## Pathogenicity tests

To verify the connection between the presence of *I. liriodendri* and the declining state of the hosts used for the isolation, pathogenicity tests were carried out on *D. virginiana* plants. After 90 days of incubation in a greenhouse, 16 out of the total 20 infected plants showed necrosis in the taproots, while mock-inoculated plants were symptomless (Fig 9 and Table 2). The incidence of symptoms was somewhat lower in the case of Ily3 and Ily5 isolates compared to the other strains. The lower pathogenicity of Ily3 and Ily5 cannot be clearly explained by the differences in exoenzyme activities (Fig 8). Both

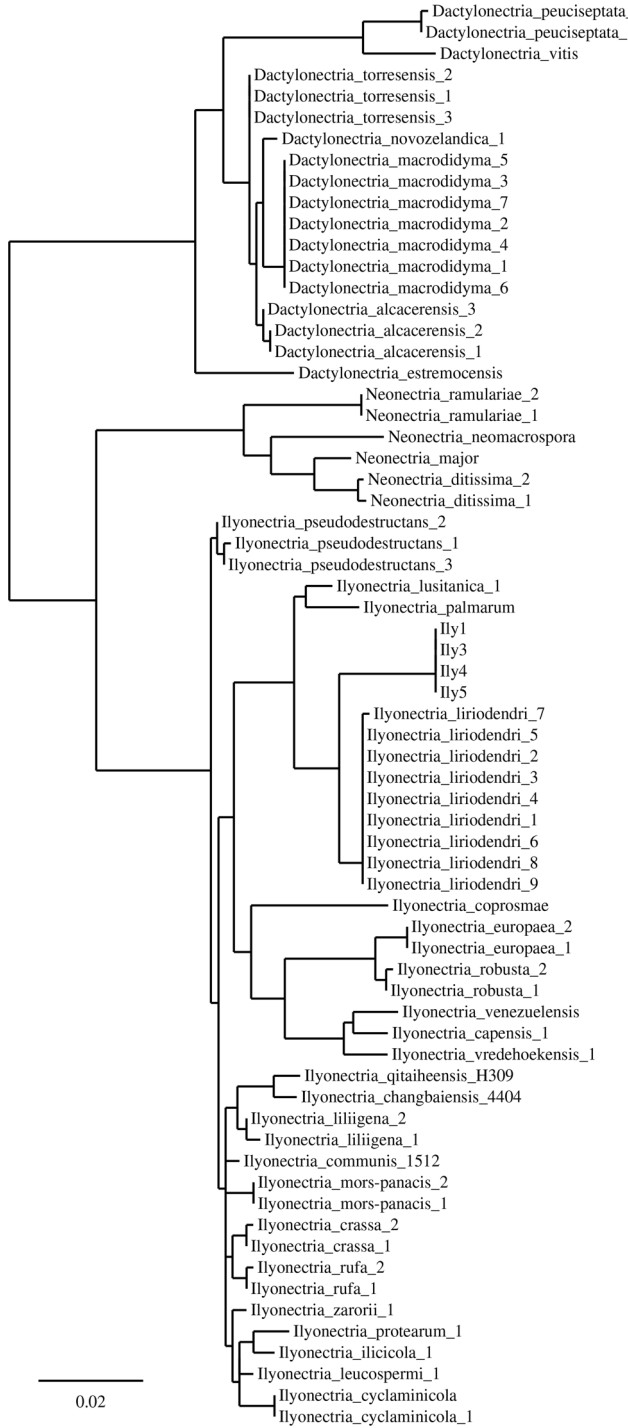

**Fig 3. Phylogenetic analysis of fungal strains.** Maximum Likelihood analysis of three genes (ITS + TUB2 + HIS) combined dataset in comparison with the four Ily isolates (Ily1, Ily3, Ily4 and Ily5). Branches with a support value below 75% are collapsed.

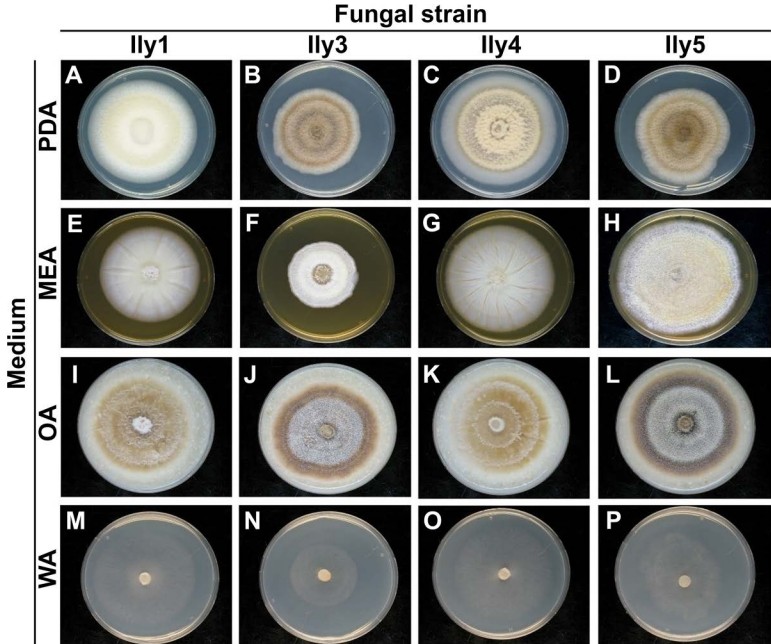

**Fig 4. Macromorphological characterization of fungal colonies.** Representative photographs *I. liriodendri* isolates (Ily1, Ily3, Ily4 and Ily5) grown for 14 days, at 25 °C, in the dark, on potato dextrose agar (PDA), malt extract agar (MEA), oatmeal agar (OA), and water agar (WA).

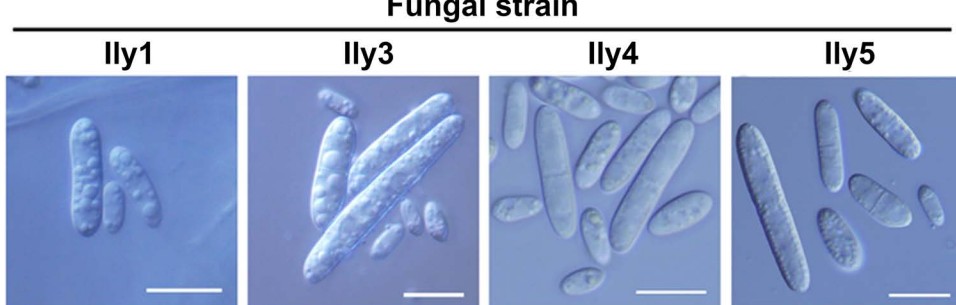

**Fig 5. Asexual spores of fungal strains.** Representative DIC photomicrographs of micro- and macroconidia of the examined *I. liriodendri* isolates (Ily1, Ily3, Ily4 and Ily5). Scalebars represent 10 μm.

strains are efficient laccase (Fig 8A), and amylase (Fig 8C) producers. Although Ily3 showed low cellulase activity (Fig 8B), it can not explain why Ily5 is less virulent compared to Ily1 and Ily4. Possible connection between enzyme activities and virulence needs further studies.

In most cases, the xylem necroses were diffuse, with central (Fig 9A) or lateral (Fig 9B) localization. Spotted necroses were detected only in three taproots (Fig 9C), out of the 16 symptomatic plants. The severity of root necroses showed high variances within a single fungal strain, hampering statistical comparisons. Contrary to the symptoms detected in artificially inoculated plants, necroses were observed in separate tracheae in the taproots of naturally infected persimmon (Fig 2). This difference is probably due to the combined effects of the excessive damage caused to the roots and the amount of the applied fungal inocula in the pathogenicity tests. The cutting of taproots and the application of high concentrations

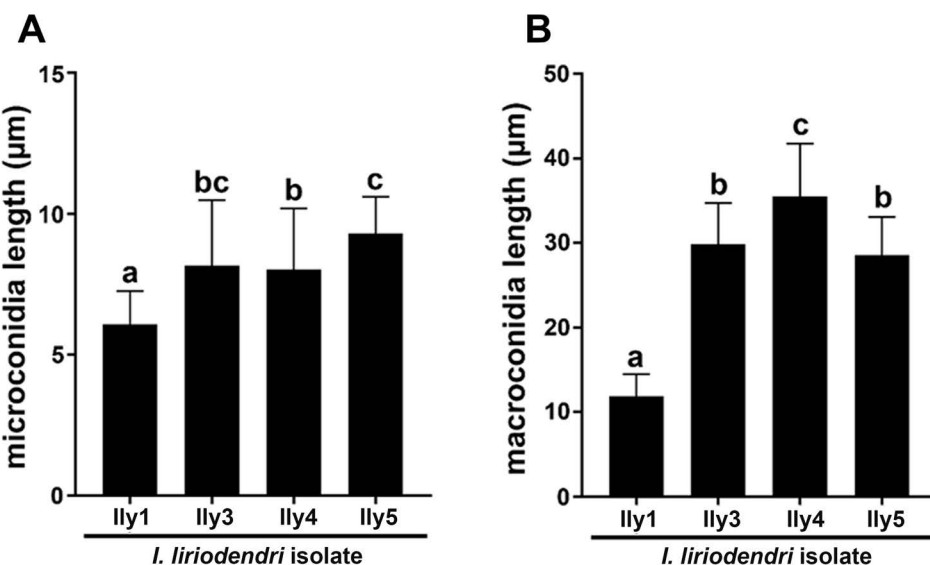

**Fig 6. Characterization of fungal strains according to conidial dimension.** Mean values and standard deviations of the length of microconidia **(A)** and macroconidia **(B)** of the examined *I. liriodendri* isolates (Ily1, Ily3, Ily4 and Ily5), grown on PDA medium, at 25 °C, for 14 days. Letters mark significantly ($p < 0.05$) differing datasets.

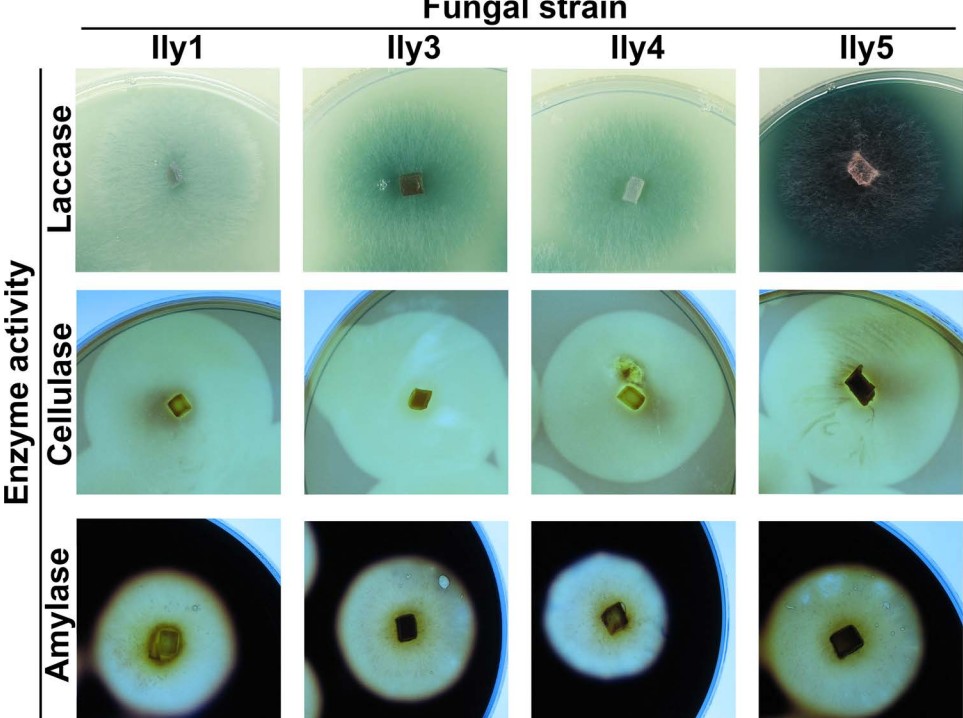

**Fig 7. Exoenzyme production of fungal strains.** Colonies of *I. liriodendri* Ily1, Ily3, Ily4 and Ily5 strains were grown on media indicating laccases **(A-D)**, cellulases **(E-H)** and amylases **(I-L)**. Photographs were taken after 7 days of incubation at 25°C.

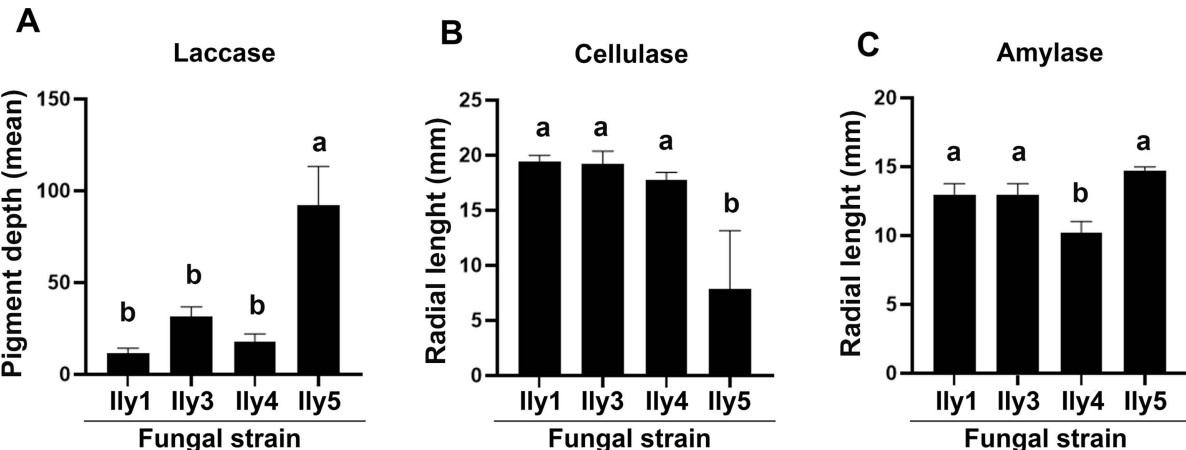

**Fig 8. Quantification of fungal strains exoenzyme production.** Laccase **(A)**, cellulase **(B)** and amylase **(C)** activities of *I. liriodendri* Ily1, Ily3, Ily4 and Ily5 strains. Different letters mark significantly differing ($p < 0.05$) datasets.

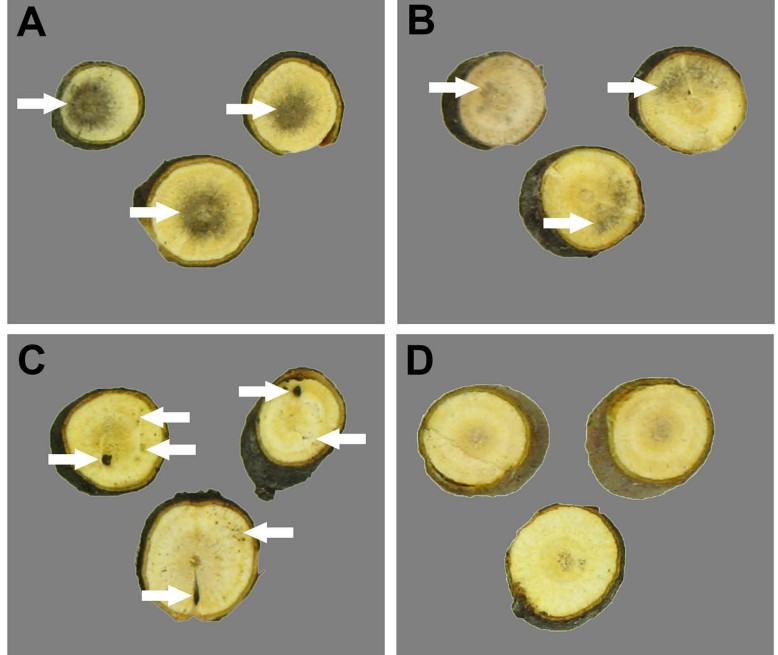

**Fig 9. Symptoms caused by the fungal strains.** Representative photographs of taproot cross sections of *D. virginiana* plants infected with *I. liriodendri* Ily1, Ily3, Ily4 and Ily5 isolates, or left uninoculated. Necroses with diffuse central **(A)**, diffuse lateral **(B)**, or spot-like **(C)** distribution were observed, while control plants remained symptomless **(D)**.

of fungal spores possibly resulted in the simultaneous infection of several tracheae, contrary to the infection of separate tracheae in naturally infected plants.

All tested strains can be reisolated frequently from the infected plants (Table 2), fulfilling the Koch postulates. The high number of causal agents, lack of specific symptoms, and the hidden nature of vascular fungal infections make it difficult to

**Table 2. Symptom incidence and reisolation rate of fungal strains in pathogenicity assays.**

|  | Isolate ID | | | |
|  | Ily1 | Ily3 | Ily4 | Ily5 |
| --- | --- | --- | --- | --- |
| symptom frequency | 5/5 | 3/5 | 5/5 | 3/5 |
| reisolation frequency | 4/5 | 4/5 | 5/5 | 5/5 |

associate novel pathogens with these syndromes. The lack of available information on the pathogenicity of the extensively studied *I. liriodendri* pathogen on the worldwide cultivated persimmon host [34] raises the possibility that the observed disease is a result of specific conditions. The possible explanation could be the cultivation of the host outside its native range. *D. virginiana* is indigenous to the southeastern United States [35], while diseased plants were obtained in Hungary. Symptoms were observed in 2021, a year affected by drought and frequent heatwaves in the previous vintage [36]. These adverse climatic conditions may weaken the plants, making them more susceptible to diseases. In addition, in Hungary persimmon plants may encounter fungal strains with which it has not co-evolved. It is also notable that the origin of the diseased *D. virginiana* plants (Eger City) is the center of a significant wine region in Hungary [37] surrounded by numerous vineyards. No literature reports on the occurrence of grapevine black foot disease in the region. However, a previous DNA metabarcoding analysis suggests that *Ilyonectria* taxon is abundant in the local soil [38]. This study used the partial ITS sequence for identification, which is not suitable for species-level identification of fungi. However, it is easily possible that *I. liriodendri* was latently present in the soil. It was previously demonstrated that soil pathogen load is associated with host susceptibility to grapevine black foot disease [39]. In addition to soil, the grapevine host itself, as well as irrigation water and cutting tools, may also serve as a source of pathogen inocula [40].

## Conclusions

This study is the first report of the fungus *I. liriodendri* in persimmon based on morphological and molecular characterization of fungal isolates. In addition, pathogenicity tests led to the conclusion that *I. liriodendri* may contribute to the development of vascular fungal infection in the persimmon species, *D. virginiana*. The putative epidemiological relevance of this finding needs to be verified by examining the susceptibility of other members of the *Diospyros* genus to the infection. In addition, the distribution of the pathogenic trait against persimmon species also needs to be studied in geographically distant *I. liriodendri* populations.

The finding that *I. liriodendri*, a black foot disease pathogen, can infect *D. virginiana* has a potential impact on the persimmon industry. General practices of persimmon propagation and cultivation should be reconsidered, and new practices should be applied. Cutting back the taproot of persimmon rootstocks to enhance the growth of auxiliary roots is a general practice during propagation [41]. This provides a suitable entry point for the soil-borne black foot disease pathogens. Protection methods can be adapted from grapevine propagation, applying hot water [42], fungicides [43], or biocontrol agents [44] on roots. The mycorrhizal fungus *Glomus intraradices* would be a suitable candidate against black foot disease in persimmon since it can colonize and enhance the growth of *D. virginiana* [45], and has been proven to be a potent antagonist of black foot disease pathogens on grapevine [46].

## Supporting information

**S1 Fig. *I. liriodendri* species-specific PCR gel electrophoresis results.** A clearly detectable amplicon at the expected size (~200 dp) in case of all four tested fungal isolates (Ily1, Ily3, Ily4, Ily5) was observed.
(TIF)

**S2 Fig. Original gel image for S1 Fig.**
(TIF)

**S1 Table. Raw data of micro- and macroconidia length of examined fungal strains in μm.**
(XLSX)

**S2 Table. Raw data of enzyme activities of examined fungal strains.**
(XLSX)

## Acknowledgments

The authors would like to thank Péter Krakkó for his cooperation with the research group by providing the infected plant material.

## Author contributions

**Conceptualization:** Zoltán Karácsony.

**Funding acquisition:** Kálmán Zoltán Váczy.

**Investigation:** Nikolett Molnár, Dóra Szabó, Adrienn Geiger.

**Supervision:** Zoltán Karácsony.

**Visualization:** Zoltán Karácsony.

**Writing – original draft:** Nikolett Molnár.

**Writing – review & editing:** József Geml, Kálmán Zoltán Váczy.

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
