## [Decision Letter · Decision Letter 0]

22 Oct 2025

Dear Dr. Zoltán,

Thank you for submitting your manuscript to PLOS ONE. After careful consideration, we feel that it has merit but does not fully meet PLOS ONE’s publication criteria as it currently stands. Therefore, we invite you to submit a revised version of the manuscript that addresses the points raised during the review process.

**ACADEMIC EDITOR:** Now both reviewers have commented about the MS, they recommended minor revision. The study is interesting and well done, however some minor revisions are need before its acceptation. Therefore i recommend that the authors follow carefully all comments done by the reviewers answering every comment.

We look forward to receiving your revised manuscript.

Kind regards,

Estibaliz Sansinenea

Academic Editor

PLOS ONE

Journal Requirements:

https://journals.plos.org/plosone/s/file?id=ba62/PLOSOne_formatting_sample_title_authors_affiliations.pdf
**.**

2. We note that your Data Availability Statement is currently as follows: [All the relevant data are presented in the paper or in the supplementary files.]

“This work was funded by the National Research, Development and Innovation Office under the project OTKA K-143453. ZK was supported by the János Bolyai Research Scholarship of the Hungarian Academy of Sciences.”

7. We note that Figure 1 in your submission contain [map/satellite] images which may be copyrighted. All PLOS content is published under the Creative Commons Attribution License (CC BY 4.0), which means that the manuscript, images, and Supporting Information files will be freely available online, and any third party is permitted to access, download, copy, distribute, and use these materials in any way, even commercially, with proper attribution. For these reasons, we cannot publish previously copyrighted maps or satellite images created using proprietary data, such as Google software (Google Maps, Street View, and Earth). For more information, see our copyright guidelines: http://journals.plos.org/plosone/s/licenses-and-copyright.

Additional Editor Comments (if provided):

Now both reviewers have commented about the MS, they recommended minor revision. The study is interesting and well done, however some minor revisions are need before its acceptation. Therefore i recommend that the authors follow carefully all comments done by the reviewers answering every comment.

Reviewers' comments:

Reviewer's Responses to Questions

**Comments to the Author**

1. Is the manuscript technically sound, and do the data support the conclusions?

Reviewer #1: Yes

Reviewer #2: Yes

2. Has the statistical analysis been performed appropriately and rigorously?

Reviewer #1: Yes

Reviewer #2: Yes

3. Have the authors made all data underlying the findings in their manuscript fully available?

Reviewer #1: Yes

Reviewer #2: Yes

4. Is the manuscript presented in an intelligible fashion and written in standard English?

Reviewer #1: Yes

Reviewer #2: Yes

Reviewer #1: Line 28. Place a "period" after I (I. liriodendri)

You mentioned that "....declining Diospyros virginiana plants," could you please attach a photo showing this "decline."

Line 64-65. Indicate the number of plants assessed and the number of infected plants.

Line 69. Indicate the brand and country of the PDA used.

Indicate how you chose the four strains in this study. How many isolates were originally obtained, and was there a randomization in choosing these four isolates?

Line 112. Provide brand, country of MEA, and OA.

Line 143. Pathogenicity tests

In your path test, did you observe a decline in the plantlets? If yes, add photos of inoculated and non-inoculated.

Add a discussion on the importance of your work in the persimon industry in Hungary.

The quality of the figures needs improvement, particularly Figure 3.

In Figures 2 and 9, please add the healthy control. Kindly place arrows indicating necrosis.

Reviewer #2: Review comments

1- The abstract presents a clear and scientifically sound summary of the study, demonstrating proper application of molecular identification and pathogenicity testing according to Koch’s postulates. However, to enhance its scientific precision, consider refining the structure and clarity of certain parts. The background could briefly emphasize the economic or ecological relevance of Ilyonectria infections. Additionally, when mentioning the phylogenetic analysis, specify the reference strains or databases used for comparison. Finally, the conclusion effectively highlights the novelty of the finding, but it could be strengthened by indicating its potential implications for disease management in Diospyros virginiana.

2- The introduction successfully builds the context for identifying Ilyonectria liriodendri as a new pathogen on persimmon. However, a few small improvements could make it stronger. The text could flow more smoothly by shortening long sentences and combining related ideas. The significance of finding I. liriodendri on Diospyros virginiana could be emphasized more clearly as a new contribution to plant pathology. Also, ensure that all scientific names are consistently

3- Material sand Methods

a. Isolation of fungal strains

Please provide more details about the number of samples collected, the conditions of incubation (duration and light/dark), and whether sterilization controls were used. This information will improve the clarity and reproducibility of the isolation method.

b. The molecular identification and phylogenetic analysis are appropriate and thorough, but please correct minor grammatical errors (e.g. its suitable → it is suitable) , adjust phrasing for consistency (e.g. Visualization of the phylogenetic tree were done → was performed), and briefly explain the need for species-specific PCR in addition to multi-locus sequencing to improve clarity and reproducibility.

c. Check the spelling of the species name (e.g. I. liriodendra → I. liriodendri) and correct the figure references (e.g., diffuse central (a), diffuse lateral (c), or spot-like (c)). It would also help to clarify why some isolates caused symptoms more frequently than others—whether this is due to differences in virulence, inoculum concentration, or plant susceptibility. Including quantitative measurements of necrosis and noting if statistical analyses were done would make the results stronger. Finally, it would be helpful to clearly explain the difference between natural and experimental infections and why this pathogen on persimmon may have been overlooked before, to make the discussion more informative.

d. Try breaking long sentences into shorter ones and separate the ideas about the host being grown outside its native range, the different climate, and exposure to unfamiliar fungal strains. It would also help to clarify the soil DNA study, noting that ITS sequencing can’t always distinguish closely related Ilyonectria species. These small changes would make the discussion more readable and strengthen your argument.

4- Results and Discussion

1) Language Comments

Line 160–161: Replace (declining D. virginiana plants) with (declining D. virginiana plants) (italicize species name).

Line 161–162: The phrase (resulting from their necrosis and/or occlusion) could be rephrased for clarity as (resulting from tissue necrosis and/or vascular occlusion).

Line 167: The phrase (isolated from the symptomatic plants) should read (isolated from symptomatic plants) (remove (the)).

Line 168: Replace (while there were slight) with (although slight differences were observed).

Line 169–170: Replace “differences between the isolates(with “differences among the isolates)

Line 170–171: Reword description of microconidia, macroconidia, and absence of chlamydospores for readability.

Line 171–174: Correct grammar and tense in the sentence about gene sequencing and multi-locus phylogenetic analysis.

Line 177–178: Fix spelling (“constracted” → “constructed”) and improve sentence structure about the phylogenetic tree.

Line 179–181: Reword PCR result sentence for clarity and consistency.

Line 266: Integrate incomplete sentence (a significant crop [31]) with the previous sentence for clarity.

Line 267: Reword (growing of the host out of its native range) → (cultivation of the host outside its native range).

Line 268: Italicize species name (D. virginiana) and improve phrasing of climatic exposure.

Line 269: Rephrase (encounters fungal strains it has not evolved to coexist with) → (encounters fungal strains with which it has not co-evolved).

Lines 270–271: Simplify sentence about hidden infections, non-specific symptoms, and multiple pathogens.

Line 272: Remove comma in (It is also notable, that…) → (It is also notable that…).

Line 273: Improve flow: (surrounded by numerous vineyards) → (which is surrounded by numerous vineyards).

Line 274: Replace (There is no literature data…) → (No literature reports…).

Lines 275–276: Clarify DNA metabarcoding sentence; italicize Ilyonectria.

Lines 277–278: Reword (which is not suitable for distinguishing black foot disease pathogens at the species level) → (which is not suitable for species-level identification of black foot disease pathogens).

2) Scientific comments

• Fungal Isolation and Morphology ( Line 168):

The phrase (similar colony morphology as described for I. liriodendri [30]) could be supported with microscopic features (e.g., conidial size).

• Micro- and Macroconidia Description (Lines 169–171):

- Providing actual measurements (in μm) for microconidia and macroconidia would strengthen the morphological identification.

- The absence of chlamydospores is mentioned, but noting whether this was consistent across all isolates would be useful.

• Strain Differentiation (Line 190):

The statement that (morphological differences…suggest that each of them represents a different strain ) is reasonable, but it would be strengthened by providing quantitative measurements such as colony diameter, hyphal characteristics, or conidial dimensions.

• Introduction to Exoenzyme Analysis (Lines 224–225):

- Consider briefly explaining the rationale for studying exoenzyme production in I. liriodendri (e.g., relevance to pathogenicity, virulence, or ecological adaptation).

• (Line 230) : Mention whether cellulase activity correlates with observed pathogenicity on host tissue.

• (Line 231) : The reduced activity in Ily4 could be discussed in the context of strain variability and potential ecological or pathogenic implications

• (Line 279): Discuss the ecological and epidemiological implications of latent soil populations serving as a local source for disease, including potential spread to nearby susceptible hosts.

5- Conclusion

• Lines (283–284):

- The statement (I. liriodendri contributes to the development of vascular fungal infection in a persimmon species, D. virginiana) is clear, but consider emphasizing whether this conclusion is based on morphological, molecular, or pathogenicity evidence.

- You could also clarify that this is the first report of I. liriodendri infecting D. virginiana, if applicable.

**Do you want your identity to be public for this peer review?** For information about this choice, including consent withdrawal, please see our Privacy Policy

Reviewer #1: No

Reviewer #2: No

---

## [Author Response · Author response to Decision Letter 1]

3 Dec 2025

Deatailed responses to reviewers are attached to the revised manuscript as separate documents.

---

## [Editor Report · Decision Letter 1]

10 Dec 2025

Isolation and characterization of the phytopathogenic fungus Ilyonectria liriodendri from persimmon as a new susceptible host

PONE-D-25-53199R1

Dear Dr. Zoltán,

We’re pleased to inform you that your manuscript has been judged scientifically suitable for publication and will be formally accepted for publication once it meets all outstanding technical requirements.

Kind regards,

Estibaliz Sansinenea

Academic Editor

PLOS One

Additional Editor Comments (optional):

The authors have followed all recommendations given by the reviewers improving the MS; therefore it can be accepted in the current form.
---

## [Editor Report · Acceptance letter]

PONE-D-25-53199R1

PLOS One

Dear Dr. Karácsony,

I'm pleased to inform you that your manuscript has been deemed suitable for publication in PLOS One. Congratulations! Your manuscript is now being handed over to our production team.

Kind regards,

on behalf of

Dr. Estibaliz Sansinenea

Academic Editor

PLOS One